# Natural Resources and the Tipping Points of Political Power—A Research Agenda

Petra Dobner 🆔 and Jasper Finkeldey *🆔

Institute of Political Science, Faculty of Philosophy 1, Martin Luther University Halle-Wittenberg, Emil-Abderhalden-Straße 26–27, 06108 Halle (Saale), Germany
* Correspondence: jasper.finkeldey@politik.uni-halle.de

**Abstract:** A general assumption concerning the relationship between natural recourses and politics is that the degradation of natural resources will destabilize political regimes causing civil wars, mass migration, or the erosion of democratic systems. Despite individual attempts to explore the relationship between different political regimes and various resources in more detail, a systematic explanation of the complex relationship between natural resources and political regimes is still lacking. In this paper, we suggest a research agenda in order to better understand their interconnectedness. We start by exploring the respective potentials of Earth system science (ESS) and the logic of Earth spheres. We argue that the notion of distinct Earth spheres has its merits but also significant disadvantages. We then propose to concentrate on a resource perspective as the more expedient starting point for investigating the nature of the interconnection between the ecosphere and humans. We refine our argument by suggesting to also reflect on the socioeconomic properties of natural resources in order to estimate their implications for political regime stability. Finally, this paper proposes three different political regime types and how each organizes its relationship vis-a-vis natural resources, especially regarding sustainable resource use.

**Keywords:** natural resources; tipping points; political power; resource regimes; earth system governance

## 1. Introduction

Earth system science (ESS) is one of the most sophisticated endeavors of natural sciences aiming at understanding the complexity and interdependencies of the different Earth spheres, i.e., understanding how our planet works and where it will head in the future. The first world model of 1972, as proposed by the Club of Rome [1], constituted a trailblazer for ESS [2]. In the past 50 years of modeling the Earth scientifically, tremendous progress has been made. At the same time, one gap persists: Despite increased efforts to integrate human action in ESS [3], it remains difficult to place human action accurately within ESS.

The urgency of filling this gap has become even more pressing with the proclamation and wide acceptance of the Anthropocene. Mankind is and "will remain a major geological force for many millennia, maybe millions of years, to come" [4]. The more important it would be to correctly determine the human impact on the Earth system and to anchor human activity as an integral and driving component of it.

To this day, the Bretherton diagram [5] p. 19 is celebrated as a major milestone on this path: For the first time, the influence of humans was integrated into the Earth system by adding a new sphere, the anthroposphere, to the Earth model. However, as a matter of fact, in this original diagram, the anthroposphere was not more than a black box at the fringes of the Earth system. As a result, the anthroposphere first came into the picture as an appendix to the natural spheres of the Earth, not as an integral part of the Earth system, let alone as the driver of change in the interconnected system of spheres. Updated versions are more complex and show the anthroposphere in more detail and depth, but the interconnections between natural spheres and anthroposphere nevertheless remain partial cf. e.g., [6].

Integrating humans in the ESS should primarily be a research agenda for social sciences. However, social sciences have never been a dominant part of ESS, and up to the present, social sciences still have not caught up with their natural science counterparts. So not only does the integration of human impact in Earth system models lacks behind [6], but also the integration of social sciences in ESS.

This diagnosis does not imply that social scientists are not interested in the progress and results of Earth models, as their vivid debate about the Anthropocene shows cf., e.g., [7–12]. The more plausible argument is that the historical natural science dominance in the field presents obstacles to an equivalent social science approach, i.e., that the way in which the field is structured scientifically does not offer easy ways to access social science contributions. The overarching question is if it is at all possible to conduct social science research adequately within the modeling framework, i.e., whether ESS in its dominant form can be a suitable basis for a social science perspective. It must be called into question whether the biogeochemical DNA of ESS is at all compatible with social dynamics. Therefore, one must ask if the dominant "confidence in formal mathematical analysis" [13] p. 284 in ESS can function as a basis for including human impact.

Some approaches try to avoid the difficulties of integrating social aspects into ESS by assuming their findings and trying to map human interaction on top. The Earth System Governance Project (ESG), for example, considers the problems outlined by ESS as a starting point. Rather than contributing to this project, ESG is dedicated to finding solutions for the conclusions of ESS, i.e., steering the Earth away from tipping points. ESG tackles this challenge by looking especially at rule systems and actor networks capable of governing the Anthropocene [14,15]. Without putting the importance of this effort into question, it remains debatable whether the ESG approach can succeed without first filling the social science gap in ESS. If human agency is the main driver of global changes, and if this driver and its dynamics are undertheorized within ESS, how then can ESG avoid that this deficiency is carried forward into global governance architectures for the Anthropocene?

One could benevolently claim that ESS and ESG are complementary projects in which one is dedicated to the natural world and the other to the social handling of it, but we believe that unless we better understand how exactly the Earth system is coupled with human impacts, we will fail to understand its rapidly changing dynamics. As long as we fail to understand these dynamics, we will not be able to find appropriate means to steer the Earth away from its tipping points.

Given the difficulties mentioned above, which point to fundamental problems regarding integrating social sciences into ESS, in this paper, we would like to propose a different approach to fill this research gap by suggesting a resource perspective that could open the door for social sciences to ESS.

First, the premise of the argument is that the human impact on Earth spheres always stems from the extraction, transformation, consumption and disposal of natural resources (NRs). Focusing on NRs instead of Earth spheres opens the perspective on the means and causes of the changing impact of human agency on the Earth and thus also opens up a view of potential governance options.

Second, we argue that NRs are not sufficiently defined by only pointing out their natural properties. The material nature is only one dimension of NRs that must be complemented by their social components, such as their substitutability and transportability, for example. These social dimensions of NRs are crucial for understanding the human use of them, because as much as their material ones they shape the criticality and political contestedness of NRs. If it is possible to determine these immaterial dimensions of natural resources, we argue, then also a connection can be drawn to questions of political power, to political regimes and ultimately also to possibilities of sustainable (or non-sustainable) governance. A resource perspective therefore may provide access to understanding human impacts on the Earth system.

Third, most social science approaches including ESG lack an understanding of the resource base for the survival of political regimes. ESG does not conceptualize NRs with

regard to political power, but is more interested in potentials and possibilities for democracy in earth system governance. We argue instead that theorizing the connection between NR and political regimes is crucial, especially for democratic regimes (see Figure 1). Political regime stability depends on the availability of resources as is demonstrated clearly in different current resource crises. Foreign policy and international development cooperation are increasingly characterized by geostrategic maneuvering to deal with growing consumption of and simultaneously declining availability of resources. By addressing the connection between NRs and political regimes we therefore understand the motives and functions of the "human" factor in ESS better. The "social boundaries approach" as suggested by Brand et al. (2021) is connected to our approach in that it looks at the socio-political structures that enable or block sustainable resource use. However, the framework does not attempt to combine geophysical and social analysis [16] p. 280.

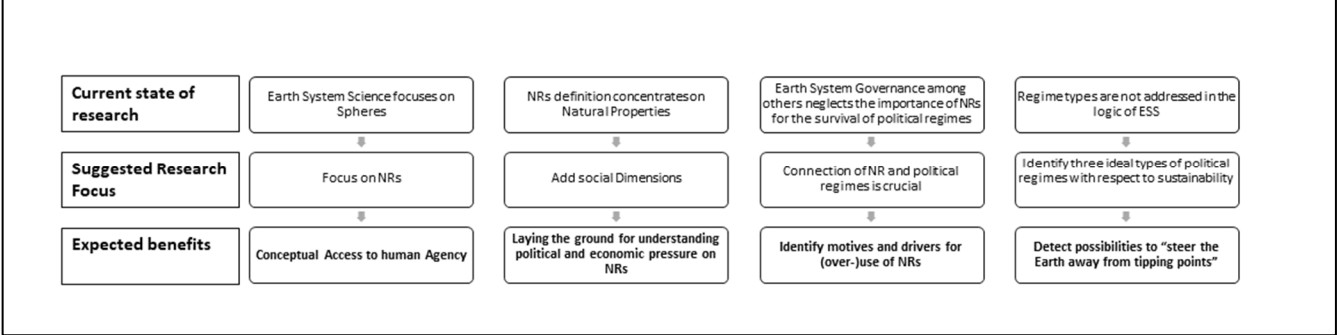

**Figure 1.** State of research and focus of this article; source: own.

Fourth, we discuss three conceivable regime types concerning human resource use. We highlight that different regime types might choose to orient themselves toward more sustainable resource uses. Political regimes thus have some leverage over the allocation and uses of resources, and they include different chances "to steer the earth away from tipping points".

Given the complexity of the subject matter and the plethora of approaches to grasping this complexity analytically, we understand our essay as a contribution to the discussion and invitation for critical examination of the suggested resource perspective.

## 2. Natural Resources

Raw materials are taken from the environment for human use and consumption and turned into resources in the process. For example, crude oil remained in the Earth's crust as a raw material for millions of years before it was unearthed to become a resource that changed global energy consumption patterns [16]. Generally, all raw materials have the potential to become a resource depending on future human needs and consumption habits. Hence only in the process of using the material we can accurately speak of a resource [17–19]. The process of "resourcification", i.e., transforming a raw material into a resource, involves a number of immaterial drivers such as social, economic and cultural processes [20] and technical, legal and financial instruments [21]. The human factor also implies a material state-shift turning the raw material into another form (e.g., from coal to steam). Combining manifest extraction and the social processes that underlie the use of resources are necessary to understand resources more comprehensively.

Where are resources situated in Earth models? In its most basic form, the Earth system is defined by its two main component parts: the ecosphere and the human factor (or anthroposphere) [22] p. 25. The ecosphere is made up of different sub-spheres, namely the atmosphere, the biosphere, the cryosphere, among others. This taxonomy clearly shows the biogeological influence in the construction of the Earth system model: spheres are named according to their composition, territorially determinable, analytically separable from each other and mutually exclusive. In contrast, the anthroposphere is "defined as the part of

the environment that is made or modified by humans. Put differently, the anthroposphere is the sphere of the earth system or its subsystems where human activities constitute a significant source of change through the use and subsequent transformation of natural resources, as well as through the deposition of waste and emissions" [23] p. 284. The anthroposphere is not a geological formation, but a sphere of human interaction entangled in all other spheres. It is not simply given, not territorially fixed and does not exist outside the Earth spheres or to put it differently, the anthroposphere is completely different from all other Earth spheres.

Moreover, according to the spheres logic, resources "travel" from one sphere to the other: They are raw materials in the geological spheres, and become resources in the anthroposphere. We would like to argue that this distinction needs to be redefined: NRs are hybrids in that they neither fully belong to the ecosphere nor the anthroposphere. At the moment when resources become resources it no longer makes sense to distinguish between natural and social spheres. The problem should be first understood on the conceptual level. The concept of the Anthropocene renders visible that human influence is planetary, so in the terminology of Earth system research, the Anthroposphere is, in fact, not an additional sphere but one that permeates all other spheres (other authors have tried to grasp this human influence differently. Moore ([24]) suggests the term 'Capitalocene' to describe the historical accumulation regime that produces 'cheap nature'. According to him, capitalistic accumulation processes are acting through nature and are responsible for destructive and exploitative practices rather than the anthroposphere as a whole. The term 'technosphere' shifts the attention to social systems that are produced by the use of technology ([25,26]). Both terms can help to understand the human impact on the Earth system. The downside of both concepts is however that they only tell a partial story. Both capitalism's exploitative relationship with nature as well as the use of technology are partial drivers changing the state of ecological systems). To put it differently: The sphere logic of Earth system science is not well suited for gaining analytical access to human impact on the planet.

In sum, we believe that the sphere terminology is flawed for mainly five reasons.

(1.) The Anthroposphere is not separate but entangled with the other spheres: The addition of the Anthroposphere to the other Earth spheres is only metaphorically convincing but actually blurs the fact that humans are not living separately in "their" sphere. This artificial separation between spheres obscures that humans put to work raw materials from the ecosphere by turning them into resources, and this is the act that alters the Earth's sphere.

(2.) The term Anthroposphere does not fully grasp that humans act to different degrees on Earth systems. Rightly, the term anthroposphere refers to the fact that humans are at all times and in all places deeply dependent on intervening in the Earth's spheres. At the same time, the term conceals that this intervention has increased exponentially in the last 250 years of industrialization and capitalist modes of production. The concept does not allow for a clear distinction between the basic dependence of human beings on the spheres of the Earth and the specific manifestation of the Anthroposphere in capitalism. Moreover, the fundamental critique of the 'Capitalocene', that not all humans are equally involved in the exploitation of the planet [9,27,28], also applies to the Anthroposphere. Unlike the term Anthroposphere implies, a resource perspective shows that not all the inhabitants of the Anthroposphere act together (they never do) but that the human factor can only be described as partial and contentious. For example, the use of coal was historically organized by a small number of industrializing Western countries that identified coal as a key resource for development. It was thus the entanglement of a small class of venturing industrialists exploiting raw materials and turning them into resources that ultimately caused qualitative and qualitative changes to both industry and nature.

(3.) The separation between different spheres is also problematic as raw materials are bundled up between different Earth spheres in reality. For example, coal extraction and use affect the hydrosphere, atmosphere as well as the lithosphere. The terminology of

the spheres, on the other hand, creates the idea of a separation that in fact does not always exist.

(4.) The separation into spheres is also questionable in that people themselves are increasingly affected by the effects of a deteriorated ecosphere. Humans, thus, can only analytically be separated from the Earth spheres they are part of.

(5.) In political terms and in relation to ESG, an important criticism is also that the sphere terminology is not suitable for a governance approach: All kinds of governance approaches need both an object and a subject that can be ruled. Spheres cannot be ruled; only humans.

We, therefore, propose to leave the sphere rhetoric aside and take a resource perspective for three reasons:

(1.) The typical way in which the radical disturbance of Earth sub-systems is described by Earth system scientists is by referring to the end of the Second World War as a "period of acceleration in the pace of change in the human subsystem, which induced significant changes in the biogeophysical subsystems and in the interactions between all such subsystems" [29] p. 382. Undoubtedly the human subsystem or Anthroposphere has major consequences for the sub-spheres of the planet and causes feedback loops between the spheres with sometimes unpredictable and often unintended side effects. While not every change in the Earth's spheres can be related to the human factor, volcanos or meteorites have changed the climate before there was any significant human impact on the climate increases in $CO_2$ concentration in the past 250 years, just to take one example, can safely be attributed to industrializations in some parts of the world. To be more precise: If mankind is to reach even more of the planetary boundaries [30] in the upcoming years, this will be the result of human action in the first place. Human production and consumption networks using natural resources were the central driver behind the great acceleration, not the Anthroposphere as such.

(2.) Spheres are not produced, consumed, or discarded; resources are. Around 100 billion tons of raw materials are excavated from the Earth crust each year, which is the equivalent of 15 tons for every citizen on the planet (and, of course, far more for those living in the industrialized zones of the world and far less for all less consumptive Earth citizens). "In only 50 years, global use of materials has nearly quadrupled—outpacing population growth. In 1972, as the Club of Rome's report Limits to Growth was published, the world consumed 28.6 billion tons. By 2000, this had gone up to 54.9 billion tons and as of 2019, it surpassed 100 billion tons. Rising waste levels are accompanying the rapid acceleration of consumption: Ultimately, over 90% of all materials extracted and used are wasted" [31] p. 9. Equally alarming is the result of this massive exploitation for technical applications. The Technosphere comprises of all the structures that humans have created: Houses, computers, armaments, and dumping sites, for example. "Preliminary estimates suggest a technosphere mass of approximately 30 trillion tons (Tt), which helps support a human biomass that, despite recent growth, is ~5 orders of magnitude smaller" [32].

(3.) A resource perspective that we are proposing here takes into view the modes of production and consumption that underlie resource uses. Coal extraction and use, for example, was never intended to negatively affect the hydrosphere, atmosphere, or lithosphere. The point is that the intention was to produce certain resources for sale and not to damage Earth sub-systems. Unless we understand the social dimensions of resources as we will further develop in the next section, we fail to understand the drivers of human impacts to the Earth system. A resource perspective thus looks at production networks *within* the Earth system as well as the effects of resource depletion for the Earth system (our plea for rethinking the terminology of the spheres should not, however, be confused with a posthumanist fusion of humanity and nature, which is now also favored as a new perspective in ESG: "This constructed dichotomy of humans and their environment, however, no longer aligns with advances in integrated

system analysis and discussions of the 'Anthropocene'. The more recent perspectives emphasize instead the complete *integration* of human and non-human agency in complex socio-ecological systems, from local scales-such as forests or water bodies-up to regional scales, such as the Alpine region, and the entire Earth system. A socio-ecological system perspective breaks down conceptual barriers between humans and their 'surroundings' and integrates them in a complex understanding where agency is diffuse, interactions are dynamic, and boundaries become blurred. [ . . . ] In short, the dichotomy of humans and 'nature', constitutive for 'environmental policy', loses its significance. Ontologically, we should no longer see humans as a distinct unit surrounded by a non-human 'natural environment', but as integral part of complex 'socio-ecological systems' at various scales, from local systems up to the Earth system. This fundamental critique of its very foundational notion of a 'natural' 'environment' challenges the 'environmental policy' paradigm ([33])." While we share the idea that humans are embedded in an environment and that mutual interactions between humans and nature play an important role in understanding of the Earth system, this should not be answered by dissolving all conceptual boundaries between humans and nature. Humans made climate change, not plants and animals, and it is human responsibility to bring about change if this is still possible).

## 3. Socioeconomic Dimensions of NRs

In their broadest sense, natural resources are specified by their usefulness for humans, i.e., social utility is the decisive criteria for determining natural resources in the first place. The Encyclopedia Americana defines NRs as "naturally occurring materials that are useful to man or could be useful under conceivable technological, economic or social circumstances" [34] p. 792. The OECD narrows the definition by stating that NRs are "natural assets (raw materials) occurring in nature that can be used for economic production or consumption" [35].

There is a clear bias toward natural science explanations of NRs. Further classifications of NRs rely almost exclusively on their geophysical properties, e.g., [36]. In addition, NRs are usually divided into (quantitatively limited) non-renewables and (unlimited) renewables (however, the notion of limited stock resources on the one hand and unlimited flow resources on the other is debatable: "[W]hen resource issues are included, traditional distinctions between renewable and nonrenewable resources lead to much confusion. Many so-called renewable resources (eroded soils, endangered species, 1000-year-old tropical forests) are not renewable in any practical sense. On the other hand, many non-renewable resources (coal, oil and certain minerals), if not inexhaustible in an absolute sense, are inexhaustible in a practical sense, because of technology, substitution and the operation of the market" [37] p. 53). Even though the social perspective of (re)usability appears in discussions on renewable and non-renewable resources, the focus is on its material origin and natural properties. Apart from some economically dominated refinements (which follow the theory of scarce or public goods in particular and aim at aspects such as exclusion from and rivalry in consumption), further subdivisions are made primarily by the natural sciences, which distinguish, for example, between mineral and non-mineral or biotic and fossil raw materials, and are based on the natural properties, not social ones.

Despite the fundamental stipulation that NRs only become resources through human use, socioeconomic determinants are almost completely absent in the further study of them. Without denying the geophysical properties to be important (so is the distribution and use of natural resources also shaped by political struggles that depend on the geophysical properties. Rivalries are very likely to occur over an abiotic resource on high demand such as oil; rivalries over water are on the rise (cf. https://www.worldwater.org/conflict/map/, accessed on 3 November 2022)), we argue that unless the socioeconomic dimensions of NRs are understood to be on par with material categories, the full implications of the human factor will remain obscured. While we do not deny that numerous disciplines have found interest in natural resources in the last decades, among them law [38] and

anthropology [39], international relations [40] and economics [41], and that all of them left their marks on the definition of NRs, an integrated perspective that links the two is yet missing: NRs differ in terms of central social categories that play an important role in their appropriation and use. The social dimensions primarily comprise of availability (including accessibility and seasonality), resource use intensity, substitutability, transportability, recyclability, excludability from and rivalry in consumption.

There are multiple possible combinations of these socioeconomic and geophysical categories. For example, if a resource is scarce globally, overused, and not substitutable, its criticality is high and political conflicts over these resources are likely (see Table 1). In terms of criticality, these different factors can reinforce or cancel each other out. This might be the case if a resource is scarce but easily substitutable. In sum, the combination of socioeconomic and geophysical properties of NRs determine their economic, social, and political impact. It follows that the drive for the appropriation of scarce resources might be fierce, which will be reflected in the section on NRs and political power.

**Table 1.** Socioeconomic and geophysical dimensions of NRs water and oil; source: own.

|  | **Water** | **Oil** |
|---|---|---|
| Biotic/abiotic | Biotic | Abiotic |
| Substitutable | No | Yes |
| Recyclable | Yes | No |
| Transportable | Yes, but difficult | Yes |
| Seasonal | Yes | No |
| Excludable | Yes. To a lower degree | Yes. To a high degree |
| Rivalries | Yes | Yes |

To provide an example of how these categories could be used, let us turn to water as a resource first. Water, a biotic resource, varies greatly globally in terms of availability. This also relates to the subcategories of accessibility and seasonality: Where water is fundamentally scarce, issues of accessibility and seasonal availability become of greater importance. Groundwater is less accessible than surface water (and often non-renewable in a manageable time horizon). Where accessibility is limited, the intensity of water use is at the same time high, especially considering that water must be used for all life purposes on Earth and is not substitutable. The transportability of water is limited: While water is often transported via canals, trucks, or even ships, the follow-up costs tend to be high, and capacities are limited. A global redistribution from regions with sufficient water resources to regions threatened by scarcity, therefore, has narrow limits. However, water is, in principle, recyclable, at least if it is used in production and not consumed and if water pollution is too severe. With regard to the central categories of public goods theory, there is a clear rivalry in consumption and the excludability from consumption depending on the specific form of water supply. All in all, water thus shows a high potential for social and economic conflict since the different social dimensions of the resource add a number of stressors to each other.

The abiotic NR oil, on the other hand, shows different features: Unlike water, oil is substitutable and much easier to transport. Its availability does not depend on seasonal aspects but varies greatly across nations and technical possibilities. It shares the rivalry in consumption with water but differs with respect to the degree of excludability.

Where do these considerations take us?

(1.) As argued in the first part, we assume that the Anthropocene situation can be described with regard to NRs as the human point of intervention in the Earth system. At the same time, this opens up a governance perspective on how humans have access to natural resources that can be politically controlled. Earth spheres cannot be governed the way NRs can be governed.

(2.) Studies of resource governance are often boxed in very narrow sectors and lack a broader understanding and comparison of different social dimensions of resources.

The mapping of social dimensions of NRs can be used to categorize NRs in sociological terms in order to focus on different degrees of dependency and criticality, on the one hand, but also on different degrees of strategic appropriation needs on the other. Moreover, adding the social dimensions also facilitates to draw a connection between political regimes, their preconditions for regime maintenance and the Earth system.

(3.) The criticality of natural resources for humans is not the same as the depletion or degradation of certain resources. Indeed, parts of the Earth's sphere might be destroyed without much harm to human needs on Earth in the short term. Only after some time, the overuse of resources might cause the arrival of tipping points beyond which environmental factors might become uncontrollable with some delay. Conversely, a number of resources are deemed critical well before these resources are depleting.

## 4. NR and Political Power

In political science and broader social science discussions, resources are dealt with in different ways: First, they are the focus of resource curse approaches that a negative effect of resource-dependent economies for equitableincome distribution, legitimacy, and stability of political regimes and development potentials [42]. Secondly, they are analyzed in international relations by looking at geopolitical power strategies [43–45]. Thirdly, they occur in an indirect way when wealth and resource abundance is depicted as an element of the effectiveness of the system, which has an influence on the legitimacy and political support of the respective regime among the population [46] (p 295). Fourthly, resources play a role in Marxist literature and the explanation of "ecological imperialism", which "presents itself most obviously in the following ways: The pillage of the resources of some countries by others and the transformation of whole ecosystems upon which states and nations depend; massive movements of population and labor that are interconnected with the extraction and transfer of resources; the exploitation of ecological vulnerabilities of societies to promote imperialist control; the dumping of ecological wastes in ways that widen the chasm between center and periphery; and overall, the creation of a global 'metabolic rift' that characterizes the relation of capitalism to the environment, and at the same time limits capitalist development." [47] p. 187. Fifth, novel discussions on financial indicators point to mixed results of the contribution of natural resource rents in relation to environmental sustainability [48]. Contrary to the fourth approach, there is a persistent belief in economic growth to invest in greener technologies [48,49].

While appreciating these approaches, we see two gaps in the discussion: First, there is a lack of systematic consideration of the direct influence of scarce resources on political regimes. So far, this has only been considered very vaguely with regard to the influence of economic effectiveness on legitimacy, i.e., scarce resources will have negative effects on the economy, and weak economic performances will have negative effects on political legitimacy. However, it can also be assumed that direct consequences for political regimes arise when resources become scarce or expensive. This is currently visible, for example, in water-scarce Iran, where the regime is liquidating environmentalists who hold the state responsible for water problems [50] and is also a threat in the current gas crisis [51]. Resources also have a recognizable and direct influence on political stability elsewhere: States are regularly accused of neglect when forests burn, rivers burst their banks, and droughts destroy crops. In addition, it must be taken into account that the ability of political regimes to adequately manage their resources depends on their overall economic situation. This is reflected, for example, in the Human Development Index (HDI). States with a higher HDI are, in principle, better equipped to manage their resources than those with a lower one. For example, Bangladesh, with an HDI of 0.661 in 2021, which is highly dependent on exports from the water-intensive and water-polluting textile industry, will find it far more difficult to pursue a sustainable water policy than, for example, China, the world's largest textile producer (HDI 0.768), which has made considerable efforts to reducing water pollution in recent years with its "10-Point Water Plan" [52]. Richer countries can score

better results in sustainable NR lifecycles, such as recycling rates or waste reduction, too. A more in-depth study of the direct effects of natural resources on political regimes is therefore urgently needed.

Secondly, there is a need to integrate social science approaches to system stability and system change on the one hand and natural science models of resource development on the other. So far, this research has taken place in separate spheres: As explained, Earth system approaches have a bias toward their natural science dominance. At the same time, models of political system stability still largely relegate ecological factors to a diffuse environment of the system. This integration is necessary for two reasons: If, on the one hand, human agency and societies are central components of the Earth system, then it is not only necessary to measure their influence on the biogeochemical Earth environment more clearly, but also the influence of the biogeochemical Earth environment on humans and societies. Integration can only be conceived as a reciprocal process, and this includes analyzing the interdependencies in the "societal natural relations" [53]. On the other hand, it should also be noted that political regimes can only exercise control over natural resources if they are not destabilized or even destroyed by the lack of resources. It is, therefore, necessary to analyze more clearly how ecological and political tipping points are connected.

The methodological points of contact for such an integration are possible as natural science and social science approaches have common roots in systems theory, whose very own concern was to work out the common properties of completely different systems [54]. Such approaches could, for example, refer to Easton's basic work on political systems and try to specify what influence resources have on diffuse and specific political system support [55–57]. They could also link up with László, who discusses the fundamental connectivity of different disciplines in his early work ("lt is not the analogy of phenomena, nor yet the identity of properties, which signifies the possibility of general system theory, but the isomorphy of invariant constructs, such as laws of development, structure, and self-maintenance, occurring in differentiated forms in the manifold realms of nature. Systems theoretical laws of organization apply to biological organisms as well as to societies. Wholeness, growth, hierarchical order, domination subordination relations, control, competition, and many others, are laws or law-like regularities of phenomena of organized complexity ranging from the level of the unicellular amoeba to the supra-organic socio-cultural ecology" ([55] p. 178) and who in later works has made some considerations on the interconnectedness of social and natural macroshifts [58,59] p. 17.

No matter where one starts: The lesson to be learned from ESS is that the development of ecological resources will have an impact on the stability of political systems. Conversely, it can be learned from political systems research that loss of legitimacy and, thus, reduced governmental capacities can occur if either social and economic performance declines or the political system has to take politically unwelcome measures in order to maintain system-relevant functions. Irrespective of the still-due, more detailed clarification of the interrelationships, one thing must be said in advance: Different resource regimes can accelerate or slow down the approach to political tipping points.

## 5. Three Ideal Types of Resource Regimes

This section extrapolates the core properties of three possible resource regimes (see Table 2). We do this by offering ideal types rather than starting from real-world examples. Max Weber proposed the study of ideal types in order to group social phenomena and get a bird's eye view that would abstract from details: "It is obtained through the one-sided enhancement of one or a few points of view and through the unification (sic!) of an abundance of diffuse and discrete, here more, there less, in places not at all, existing individual phenomena, which conform to those one-sidedly emphasized points of view, into a unified conceptual image. In its conceptual purity, this conceptual image is nowhere to be found empirically in reality." [60] (p. 65 ff.; translated by the authors). The aim of ideal types is to render data more "accessible and manageable" [61] p. 5. In reality, we

find some insights from every paradigm informing contemporary resource regimes. The different resource regimes are marked by diverging approaches to what a resource actually is and have direct consequences for how the respective regime treats the environment.

**Table 2.** Three ideal resource types, source: own.

| | **Global Extractivism** | **Mitigation and Adaptation** | **Sustainable Resource Regime** |
|---|---|---|---|
| Core idea | Nature as resource to increase profits | Adapting to and mitigating risks from overexploitation of resources | Reversing extractivist logic and care for the environment |
| Worldview | Homo oeconomicus | Environmental managerialism and innovation | Gaia |
| Main schools of thought | Capitalism, market radicalism | Ecological modernization, environmental management, environmental economics, climate governance | Deep environmentalism, de-growth, environmental justice, *buen vivir* |
| Legal regime | "Permanent sovereignty over natural resources", free market oriented, deregulating | Governance, efficiency-, goal-, output-oriented, politicized | Earth system law |
| Consequence for approaching tipping points | Destruction of life-supporting systems and neglecting tipping points | Stabilizing Earth systems, managing environmental decline in the face of approaching tipping points | Reversal of great acceleration; full awareness of tipping points |

*5.1. Global Extractivism Is Marked by Exponential Resource Use and Destabilizing Effects on the Earth's Systems*

Global extractivism disregards the carrying capacity of the planet, which makes it unsustainable in the strict sense of the word and will likely destroy life on Earth [62].

To speak of natural resources already implies an extractivist mindset [39] p. 1. It means that, in principle, everything that can be found in "nature" could potentially become a "resource" for human production and consumption. Capitalism, in its distilled and untamed form, is the prime example of the extractivist logic. Under capitalism, resources are rendered commodities and put in circulation in order to generate surpluses. While the driver of capitalism is to increase capital, not to overuse resources, the by-product of ever-expanding cycles of increasing capital is the greater use of resources in the production cycle. In Marx's thinking, the only way the value of a commodity can increase is by putting human labor to work. Global extractivism is marked by the overexploitation of both human labor power and natural resources.

A global extractivist regime will disregard the carrying capacity of the planet and its resources. Large polluting corporations will try to avoid expensive pollution control and lobby against tighter regulations. The underlying worldview is that of the homo oeconomicus, who always calculates and cuts all unnecessary costs. In order to save costs, natural resources will first be extracted where resources are readily exploitable at the lowest possible costs; consequential costs are externalized and/or postponed. As capitalism is marked by competition, global extractivism is marked by a global rush to secure resources. This competition under global extractivism can take violent forms where access to resources is contested.

World extractivism is not centrally managed, but enabled by the extractivist state and its coercive apparatuses, multinational corporations (MNCs), and global finance. The driving forces of global extractivism are Northern core countries and emerging economies such as the BRICS, where enough capital is available. The state under global extractivism takes different roles but never stands in the way of extraction. The state invests in, legislates for, or owns extractive industries. The legal regime of global extractivism is therefore dominated by the idea of "'permanent sovereignty over natural resources' (PSNR)" [63] p. 155, deregulation agendas, and it will protect the freedom of the market by all legal means.

Generally, however, nation states help enable global market competition to facilitate access to resources and take different roles as watchmen over the competition, financiers, and lenders of last resort during global resource recessions.

### 5.2. Mitigation and Adaptation Are Incremental Governance Approaches toward Moderately Lowering Resource Uses While Preparing for the Impacts of the Earth's Collapsing Systems

Mitigation means finding remedies to protect the Earth's system from collapse. Resource use will be gradually lowered to avoid overexploitation or imbalances in the Earth's system. Adaptation means to plan for changes in Earth systems and anticipate shifts in Earth system patterns. Measures to adapt include flood walls and early warning systems for rapidly changing weather patterns, among others.

The main paradigms that inform mitigation and adaptation are ecological modernization and environmental management [64]. Ecological modernization proposes that modernity and environmental conservation are reconcilable. The main proposition of ecological modernization is that the environment might even be augmented for human consumption without destroying the planet. Genetically modified (GM) crops and geo-engineering have the potential to render the environment more hospitable and convenient for humans. Ultimately, proponents of mitigation and adaptation believe that the environment is manageable and might be organized such that there might even be a positive-sum game and ideal allocation of uses of resources and protection from harmful consequences.

Because of the complexity of Earth systems, there is no guarantee that mitigation and adaptation measures are sufficient. Rather than bringing extractivism to an end, mitigation and adaptation provide the means to potentially protect the Earth system from severe rebound effects and disasters. Mitigation and adaptation are thus aimed at balancing the market-driven logic of the capitalist world economy on the one hand and the sustainable stewardship of natural resources on the other. The pendulum between those two demands is always in motion. Mitigation and adaptation measures to answer environmental entropy but ultimately manage decline. The legal regime is efficiency-, goal-, and output-oriented, more focused on governance than legal formalism [65] p. 24, and subject to varying demands in the jurisdiction to support either "business as usual" or the demands for greater sustainability. The law is thus politicized and forced to innovate insofar as existing legal norms must be applied to new cases (c.f., e.g., climate litigation).

To sum up, mitigation and adaptation do not break with the extractivist logic but explore remedies for negative externalities produced by it. However, the regime has some innovative potential, especially with regard to legal procedures.

### 5.3. A Sustainable Transnational Resource Regime Is Linked to Radically Lowering Global Resource Use to Actively Steer Earth Systems Away from Tipping Points

A resource regime that is renewable in the literal sense of the word is able to reproduce itself over generations. A truly sustainable transnational resource regime is informed by paradigms such as deep environmentalism, de-growth, environmental justice, Anthropocene studies, circular economy, and related theories and paradigms [66]. The starting point is the question of how to conserve the planet as a whole through practices of rewildering, reforestation, reduction in global working hours, and partial de-industrialization in polluting sectors such as energy from fossil fuels. A sustainable resource regime aims at reversing extractivist tendencies and replacing these with sustainable practices and ethics of ecological healing.

The mode of living under a sustainable resource regime uses different measures to substitute polluting resources such as fossil fuels or regulate polluting practices. From an accounting perspective, a sustainable resource regime looks at the extraction of a non-renewable resource not as a profit but as a loss. Conventional accountancy measures such as GDP are thus replaced by ideas such as *buen vivir* [67]. A sustainable resource regime knows no negative environmental externalities that will not be accounted for. The regime will not allow free riders that do not compensate for the damages they cause is deeply embedded in global production chains.

A sustainable resource regime is anti-accelerationist and tries to reverse the "Great Acceleration" of resource use starting after World War II [68]. Much like slow cooking or walking instead of taking the car, speed is no longer a core paradigm of global production and consumption. This saves resources; less human labor power is expended, and less natural resources are consumed. Global imperatives are deduced from the danger of tipping points rather than from market imperatives such as free trade. Environmental precaution rather than taking risks informs decision making. The state takes a different role vis-à-vis resource markets than under extractivism. The state either prohibits resource markets entirely or regulates them in line with planetary boundaries. Hornborg [27] suggests that the drive to destroy the planet can only be realized by abolishing money as a key incentive superstructure underwriting extractivist logic. States, therefore, ensure that planetary life support mechanisms are kept intact insofar as they help to radically change the governance of resources. This is also reflected in a new type of legal regime, the Earth system law, which is "more sensitive to and reflective of the functioning of the earth system as a socio-ecological system" [69] p. 2. Legal norms in a sustainable resource regime are taking the precautionary principle seriously with respect to environmental and human health.

## 6. Conclusions

In this article, we are proposing a resource perspective to better understand the human factor in Earth system models as well as the link between resource use and political power. We argue that political power will more and more depend on regimes that ensure access and sustainable use of NRs. As natural support systems are approaching tipping points, so do political systems that depend on natural services. The agenda complements the existing approaches that aim to add a social dimension to the study of Earth systems. With regard to the ESG project, the NR approach fills a gap as the material underpinnings of Earth governance were undertheorized. The "social boundaries" approach is more concerned with negotiation processes between critical stakeholders and engages less with the interface between natural and social systems that we are concerned with. In this article, we suggested a four-step study program.

First, we suggest that the terminology of Earth spheres in Earth System Sciences obfuscates human interventions. Rather than the Anthroposphere acting on the biosphere in abstract terms, we argue that the point of intervention is the extraction and use of resources. NRs are hybrids no longer strictly belonging to spheres. However, this should not blur the distinction between human action and nature.

Second, we argue for considering natural resources not only in their geophysical properties but also in terms of their social dimensions. Social dimensions open the perspective on different degrees of dependency and criticality on the one hand but also on different degrees of strategic appropriation needs on the other.

Third, we point out that the relationship between NRs and political regimes needs to be more precisely defined. Two major gaps exist in our view: A lack of systematic consideration of the direct influence of scarce resources on political regimes and better integration of social science approaches to system stability and system change on the one hand and natural science models of resource development on the other.

Four, ideal types of resource governance can be useful for understanding different uses and extraction of resources. Empirically one finds features of all three ideal types in global resource governance. Extractivist logic is hegemonic globally. The political econ-

omy of world trade is promoted by the World Trade Organization (WTO), World Bank, International Monetary Fund (IMF), core capitalist states, and multinational corporations (MNCs) [70]. World society is shaped by uneven development and differentiated spaces. Hence power relations are not the same everywhere. Hegemony is never universal but subject to contestation [71]. The exploitative relationship between nature and resources is evident in most places but also resisted in some pockets around the globe [28]. The managerial paradigm is sustained by the United Nations Environmental Program (UNEP) [72]. Since the Stockholm conference in 1972, the body has put forward comprehensive policy proposals that mostly result in non-binding norms. While the UNEP calls for sustainable resource use, de facto resource use has risen exponentially since the publication of Limits to Growth [1].

At the beginning of this paper, we pointed out that we understand our contribution as a proposal for a research agenda. An overarching problem is the immense intellectual challenge posed by the Anthropocene. The realization that the Earth may be entering a final phase (for humans) and that this is due to limitless human exploitation of natural resources inevitably also leads to the realization that this has occurred on scientific, economic, political, and legal foundations that must be changed. Any effort to lay the intellectual and political grounds to better equip humanity to steer away from tipping points is welcome in this light. We see our contribution as part of this effort, which needs to be completed in further research.

In addition to the fundamental question of whether the proposed resource perspective is viable in order to better understand the impact of human activity on the Earth and to better manage it in the interests of sustainability, we see a need for research in at least three respects: First, we have pointed out that sphere rhetoric creates blind spots with regard to the Anthroposphere. While it is a step forward in integrating human action into Earth system models, the addition of an Anthroposphere to the geological spheres does not seem to be the best analytical starting point for estimating human influence on the Earth. Moreover, even if we present an alternative proposal with the resource perspective, conceptual questions remain open: How can the work of the ESS remain fruitful from a socio-political perspective? How can other connections to the ESS be conceived? For this step, an interdisciplinary discussion of the conceptual foundations of the scientific understanding of the Anthropocene is imperative.

Second, the socioeconomic dimensions of NRs need to be critically examined and systematized, and their influence on the stability of political regimes sharpened. Studies that analyze concrete situations of how political regimes respond to resource scarcity and studies of how different social dimensions of resources affect the stability of political regimes are important in this regard.

Thirdly, it needs to be clarified to what extent the proposed ideal types can, in fact, be used to steer the world away from tipping points. We see here the importance of case studies of those countries that constitutionally enshrine and politically strive for a reversal in their resource use. Under what conditions is it possible to change direction in a world that still relies on "business as usual", even though all the scientific evidence shows that there can be no "business as usual" approach? The latter is of particular importance, as only the last-mentioned ideal type is sustainable; this is at the same time the least likely in realpolitik terms, despite all the signs of crisis.

**Author Contributions:** Conceptualization, P.D. and J.F.; methodology, P.D. and J.F.; software, P.D. and J.F.; validation, P.D. and J.F.; formal analysis, P.D. and J.F.; investigation, P.D. and J.F.; resources, P.D. and J.F.; data curation, P.D. and J.F.; writing—original draft preparation, P.D. and J.F.; writing—review and editing, P.D. and J.F.; visualization, P.D. and J.F.; supervision, P.D. and J.F.; project administration, P.D. and J.F.; funding acquisition, P.D. and J.F. All authors have read and agreed to the published version of the manuscript.

**Funding:** This research was funded by the Open Access Publication Fund of the Martin-Luther-University Halle-Wittenberg.

**Institutional Review Board Statement:** Not applicable.

**Informed Consent Statement:** Not applicable.

**Data Availability Statement:** Not applicable.

**Conflicts of Interest:** The authors declare no conflict of interest.

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
