# Peer review of "Natural Resources and the Tipping Points of Political Power—A Research Agenda"

_sustainability, doi:10.3390/su142214721_

Round 1
Reviewer 1 Report
Dobner et al. submitted the manuscript "Natural Resources and the Tipping Points of Political Power – a Research Agenda" which unwinds various aspects of the utilization of natural resources and reflects future consequences.
The quality of the paper is at its core, where the authors emphasize the importance of resource utilization.
However, I have some minor remarks which are subjective:
I agree with the author's perspective on Resource utilization. But I also want to point out that, there are intergovernmental policies at the global level, which are neither synchronism nor follow any alternative policy. Instead, governments make plans based on their country's requirements or, more specifically, which address the needs of most native people. However, countries with high human development index (HDI) are equipped with better policies than underdeveloped countries with lower HDI. Therefore, the differential use of resources may depend on many factors, but to an extent, on the state of people living in those countries. For example, the major textile manufacturer are located in Bangladesh and exports a significant proportion of textiles to HDI countries (especially countries from the continent of North America and Europe continent). Therefore, amending policies governed in HDI countries cannot be equally applicable to lower HDI countries because the priorities of people living in those countries are different than those of developed countries (or countries with high HDI).
Lacking Technology or Techno-economic assessment: Although the scientific world is pushing hard to implement sustainable processes, zero waste, 100% recycling, and efficient waste management, but bringing these qualities/ properties/functionality in a commercial product significantly increases the overall cost. However, we are still struggling to bring newer technologies that produce similar production rates in a similar price range. So far, undoubtedly, we have managed to achieve some successes, but we still have a long way to go. For example, Textile industries are second-world water polluting industries because the maximum proportion of dye chemicals remain unreactive to a textile fabric and remains in water which later causes water pollution (as colored water or dye water). While one could easily argue that this can be improved with newer dyeing processes where lower boiling solvents can be evaporated and reused for iterative cycles. Unfortunately, water is so cheap that it could be replaced with a low boiling solvent, making it highly expensive that it would not be viable for commercial industries to adopt it. Secondly, evaporation requires energy, making it more expensive (2700 times more than a typical dyeing process). Therefore, alternatively, large-scale industries are cautious about their life cycle assessment of products or recyclability. But, bringing newer products with these properties even at higher prices does not guarantee the same material properties as current commercial products; for example, the hydrophobicity character of biodegradable plastic still is questionable in a broader way.
I can understand that authors can't enclose all the points, but in my opinion, adding a few lines about the points mentioned improves the paper's broadness.
I found the paper quite interesting with quite high readability, however, there are some typographical errors and formatting issues that be resolved in the revised version.
Author Response
We thank the reviewer for the important and insightful points raised in response to our paper as well as the suggestions to improve our work. We were able to include key points made in the review especially with regard to differences in individual countries’ capacities in dealing with natural resource management.
Point 1:
The reviewer’s points on countries’ state of development and their trading relations vis-à-vis other countries are key. Capabilities to manage natural resources sustainably vary from country to country. First, we respond to this point by adding the reviewer’s suggestion on the Human Development Index (HDI) highlighting that richer countries are better placed to manage natural resources more sustainably. To make this point clearer we added the examples of Bangladesh and China with respect to water consumption. A big part of Bangladesh’s excessive water consumption and pollution is related to its textile industry (which is very export-oriented). China, the world’s largest textile producer, has better access to technology to mitigate for water pollution. This is also the case for recycling and waste reduction as added to the paper as well. However, global levels of recycling and waste reduction are insufficient across nearly all countries and regions on the planet.
Point 2:
In order to strengthen our approach, we have also added a legal dimension to our paper. The three resource approaches are now complemented with considerations on legal practices within the respective regime.
Point 3:
Concerning the typos and formatting we have attempted to revise the paper to make it more readable.
Reviewer 2 Report
Dear authors,
I have the following observations, questions, and comments that may help to improve your work. The authors must modify the following points in great detail.
1. In the abstract, please include 2-3 special quantitative achievements from the findings of this study in the context of the environment by combining the research objectives and problems. Please limit your abstract to 250 words. Check spellings for many words that are misspelt or written in haste.
2. The introduction section needs a few more sentences to strengthen the article, and please include the research problem, objective, and novelty in the last paragraph of the Introduction section.
3. Include a few more sentences at the beginning of the introduction explaining your paper's contribution to how strong political will and effective foreign policy impacted on natural resources .
4. Please also present the methodology section in a concise graphical format.
5. The literature review section is very weak; please revise it.
6. Please present your literature review in the form of a SmartArt chart.
7. Just after the Methodology, please mention the societal benefits of your research in terms of evaluating its key determinant.
8. In 500-750 words, explain research problems, solutions, and the theoretical contribution of your study in the "Results" section.
9. Please include graphical presentations of your findings.
10. Describe why you placed this study in a separate section of "Policy Suggestions" just before the section of "Conclusions."
I found that the literature section is a little weak, shift your study a little more towards regime change and natural resources accessibility in different countries, therefore it requires more studies to be reviewed therefore I suggest you to include the following work:
https://doi.org/10.1016/j.resourpol.2022.102593
https://doi.org/10.1016/j.resourpol.2022.102881
I think above all studies will make this study more relevant in bridging the gap with literature.
Looking forward for your revised submission.
Author Response
We thank the reviewer for the observations, questions and comments to improve our work. We have seriously discussed each suggestion, included those we agree with, and think our article has benefited greatly. But before responding to each point in detail, we would like to make a preliminary remark: The fundamental differences between quantitative and qualitative sciences are well known. We consider these different approaches to be equally important.
Nevertheless, in this paper we work primarily qualitatively and conceptually, i.e., we are concerned with conceptual and theoretical questions. We explicitly propose a research agenda and do not present research results in the form of quantitative data. Our conceptual approach we try to overcome the inherent epistemological limits of the prevailing paradigms. The reviewer's core critique, on the other hand, calls for quantitative results and thus follows a different research tradition. Therefore, there is unfortunately a mismatch between our research agenda and the reviewers demands, that we cannot completely resolve.
However, there are some differences in quantitative and qualitative approaches that are apparent from the review. The reviewer wants us to provide quantitative achievements that are not the approach we are taking. Our paper offers first considerations on a research agenda from a conceptual perspective. While we draw on both qualitative and quantitative studies as can be seen in the references, our inquiry is not concerned with generating new quantitative data.
We answer the comments in the order presented to us.
1.
- This is no quantitative study but a conceptual paper drawing on secondary literature including both quantitative and quantitative findings. In this paper we are proposing a conceptual approach rather than presenting our own empirical research.
- We think that a conceptual shift from anthroposphere to resource-centred approaches might also be a good point of entry for quantitative scholars. For example, we take findings from studies on resource depletion and the resource curse seriously that are suggesting an inverted relationship between economic development and natural resource abundance.
2.
The third point at the end of the Introduction has been re-written to include evidence on how our approach is different from the Earth System Governance (ESG) approach as well as the “social boundaries approach”. The novelty of the approach is described in points one to four at the end of the article.
3.
In this paper we suggest that resource use and management is a function of the way we understand our relationship to natural resources; i.e. if regimes adopt an extractivist mindset or sustainable way of thinking about natural resources. Political will can take three different forms as presented with regards to the three different resource regime types. Equally, it depends on the socio-economic properties of the resource itself. The capacity of political regimes to manage natural resources sustainably varies. Countries with a higher Human Development Index (HDI) have more financial muscle to manage their natural resources sustainably.
We added a sentence on the growing importance of geostrategic manoeuvres as resources are becoming scarce.
4.
We added a graphical format of our approach (cf. Fig. I.)
5.
Instead of one single review section, we are reviewing literature in the first sections after the Introduction. Our approach is a synthesis of different existing approaches.
We thus develop a systematic approach to study natural resources regimes. This is also presented in a newly inserted figure 1.6.
The paper reviews literature in the sections on natural resources, socio-economic dimensions of natural resources as well as natural resources and political power. Our paper is organised such that there is no single literature review section, but an engagement with existing studies throughout the first mentioned sections. Of particular importance are works from the Earth System Sciences (ESS), Earth System Governance (ESG) as well as system theorists in Political Science; the literature is considered in the respective sections. Considering this, a SmartArt chart does not seem useful.
7.
We do not have a “key determinant”, but we spell out a multifaceted problem. Nevertheless, the societal benefits are spelled out in a new section we added at the end of the paper.
8.
The contributions are listed in the conclusion.
9.
We visualized our approach graphically in the beginning of the paper. As pointed out we suggest a research agenda, and we are dedicated to follow up on our research agenda. Drawing on this approach, we will be able to will write empirical papers drawing on this preliminary work.
10.
We thank the reviewer for the suggested literature that we included in our paper. We see that financialization and resource rents are one other aspect of how resources can be studied.
We were also discussing the question of regime changes.
In our paper, we are proposing to present ideal types. In reality, resource regimes are a mix of the above. It will need empirical analysis to look at how regimes change from one to the other. The literature on regime change is very focused on change of belief systems or physical force. There is a lack of studies looking at regime changes from the perspective of access to resources. In our contribution we say that more and more regimes will (have to) change because of a mix of (external) factors such as depletion of some critical resources, unequal distribution of resources and climate change due to overuse of hydrocarbons, among other resources. The governance capacities of countries to change their relationship to natural resources varies greatly. Equally, the baseline biocapacity of countries vary significantly (Alfalih & Bel Hadj 2022, p. 2)
Reviewer 3 Report
The authors present a research agenda, which aims at identifying and classifying political systems and associated resource regimes, with critical biogeochemical processes that sustain life on Earth and are key to maintaining a safe operating space for humankind (as per the Planetary Boundaries Framework). To this end, the authors rely on a critical acceptance of the premises of Earth System Science (ESS). However, they also reproach the excessive weight of natural science approaches in ESS, at the expense of critical social science approaches. They claim that this imbalance leads to a somewhat dysfunctional terminology and framinings in ESS (especially the notion of 'anthroposphere'), which distort the concrete understanding of human interference with biogeochemical processes that are key to keeping life on Earth within a safe operating space.
In order to address this shortcoming, the manuscript proposes to focus more intensely on the study of the social dimensions of natural resources, in addition to the classic focus on their geophysical properties. Such a focus on the social implications of the global flows of natural resources would allow, in view of the authors, a more systematic appraisal of the relationship between the global metabolism of natural resources and political regimes. From a methodological perspective, they argue that this is a sound approach, given the shared roots in systems theory of the underlying social science and natural science approaches. In a further step of their argument, the authors ultimately propose to investigate the performance of specific political systems against the backdrop of their relative position within the global socio-ecological metabolism. To that end, they introduce a distinction between three models or ideal types of resource regimes: (1) global extractivism based on radical market ideology, (2) mitigation and adaptation, based on ecological modernisation and environmental managerialism, and (3) sustainable resource regime, based on counter-hegemonic onto-epistemologies of deep environmentalism and de-growth. In my view, the manuscript lies down a solid ground for a future research agenda. In particular, the models or ideal types of resource regimes seem justified and have an appropriate explanatory potential to address the overall research question, i.e. assessing concrete and measurable interference of human societies with biogeochemical processes in the Earth system and steering them away from the tipping points identified in the Planetary Boundaries Framework. RECOMMENDATION 1: However, the depth and explanatory potential of ideal types would be increased if their legal dimension (underlying regulatory paradigms) were also taken into account, given the stabilising role that legal system play vis-à-vis social and political systems (e.g., by way of example, property rights and sovereignty over natural resources in ideal type 1, common concern of humankind, managerialism and procedural rights in type 2, and legal pluralism/ecumenism and collective [economic, social and cultural] rights in type 3). This would require expanding the respective sections in pp. 11 to 13, as well as reviewing the explanatory table in lines 444 & 445. To this end, again only by way of mere suggestions, the works of Poul F Kjaer on the Law of Political Economy, Marie-Catherine Petersmann et al on the constitutionalisation of the Anthropocene, and Rakhyun Kim on Earth System Law would seem appropriate points of reference for addressing the challenges of global governance, sustainability and the Earth system from the perspective of global/transnational law. RECOMMENDATION 2: Finally, I also think that the overall presentation of this research agenda should be spelled out in some more detail. At present, the conclusions of the manuscript read as a summary of the prior points, but could be redrafted in order to explain more clearly how the authors´research agenda relates to and, potentially, also complements or adds to concurrent, currently ongoing, research endeavours. So, for instance, it would gain sharpness if the following aspects were addressed:- Outline more clearly, especially in the conclusions (pp. 13 & 14), how this agenda relates with, deviates from, and/or complements the (competing) research agenda in the framework of the Earth System Governance project (especially its latest implementation plan).
- Closely related to the previous recommendation, the conclusions should also explain more openly how the authors´agenda relates with the research agenda of the Societal Boundaries Framework proposed by Brand et al. in 2021.
Author Response
We thank the reviewer for the important and insightful points raised in response to our paper as well as the suggestions to improve our work. We were able to include key points made in the review especially with regard to legal dimension of natural resource regimes and the similarities/differences to the Earth System Governance Implementation Plan as well as the social boundaries approach.
First, in order to appreciate the legal dimensions of the three proposed resource regimes, we added a legal dimension to the respective approaches using the work of Kjaer and others. The reviewer’s comments also inspired us to add an extra row on the respective legal regime underpinning the respective resource regimes.
To the introduction and conclusion we added some clarifications on the similarities and differences between our approach and the ESG and social boundaries approaches. Namely the ESG Implementation Plan is not concerned with the relationship between natural resources and power. We were trying to fill this gap in our contribution. The social boundaries approach by Brand et al. (2021) is more concerned with social negotiation processes and power relations between key stakeholders and is less concerned with categorizing and systematizing the interaction of social and natural systems as we have attempted. Therefore, our work is in productive exchange with the two former approaches.
Regarding the potential lack of empirical examples to illustrate our case, we added elaborations on the Human Development Index (HDI) and the capacities of countries to manage their natural resources sustainably. In essence, countries with higher HDIs are in a better position to engage in more sustainable resource management. However, global levels of recycling and waste reduction are insufficient across nearly all countries and regions on the planet.